# Various Endoscopic Approaches for Removal of Proventricular Foreign Bodies in Parrots—Three Case Reports

**DOI:** 10.3390/ani13243839

**Published:** 2023-12-13

**Authors:** Sungryong Kim, Nari Kim, Hakhyun Kim, Ki-Jeong Na, Eui-Ju Hong, Dong-Hyuk Jeong

**Affiliations:** 1College of Veterinary Medicine, Chungbuk National University, Cheongju 28644, Republic of Korea; vet08dannyk@gmail.com (S.K.); queng91@gmail.com (N.K.); kimhakhyun@cbnu.ac.kr (H.K.); sigol@cbnu.ac.kr (K.-J.N.); 2College of Veterinary Medicine, Chungnam National University, Daejeon 34134, Republic of Korea; ejhong@cnu.ac.kr

**Keywords:** psittacine birds, gastrointestinal foreign body, flexible endoscopy, non-invasive operation, anesthesia method

## Abstract

**Simple Summary:**

This report describes three successful cases of gastrointestinal foreign body removal from psittacine birds via endoscopy. Our findings suggest that a potentially effective endoscopic removal approach may involve ventral recumbency with anesthesia via a mask or endotracheal intubation when dealing with a limited number of blunt foreign bodies within a short operation time. This case report offers valuable insights and practical guidance for the efficient and safe removal of foreign bodies in avian patients.

**Abstract:**

Although the use of incision-free endoscopy for foreign body (FB) removal in dogs and cats has been extensively documented, its application in birds remains limited. Thus, we present the endoscopic removal of gastrointestinal (GI) FBs from psittacine birds, employing different patient positioning and anesthesia methods. Two blue-and-yellow macaws (*Ara ararauna*) and a Triton cockatoo (*Cacatua galerita triton*) were examined. X-ray imaging revealed FBs situated in the proventriculus in each case. The FBs, all identified as feeding tubes, were safely removed using grasping forceps during the endoscopic procedure, and no severe complications occurred. Based on the outcomes of each operation, the most suitable patient position may be ventral recumbency rather than dorsal recumbency, with the use of a mask or endotracheal intubation, depending on the anticipated operation time. However, a larger number of cases would be necessary to confirm the optimal patient positioning and anesthesia method.

## 1. Introduction

Gastrointestinal (GI) foreign bodies (FBs) have been reported in diverse avian species, especially in pet and captive birds. Commonly encountered FBs include items such as feeding tubes [1], wooden perches [2], wire [3,4], and fibrous materials, such as tent-like huts [5] and artificial grass fibers [6].

Juvenile psittacine birds are usually curious and frequently use their beaks to play with several items in their environment. Specifically, juvenile umbrella cockatoos and African gray parrots may exhibit behavioral propensity to ingest foreign objects [2]. Consequently, various cage substrates and household items have been discovered in the crop [1,7], proventriculus [2,3,4,5,8], ventriculus [4,6,8], and intestines [9] of pet and captive birds.

Birds that ingest FBs normally exhibit vague symptoms or are asymptomatic; however, some birds exhibit lethargy, weight loss, regurgitation [8], and paresis [6]. Differential diagnosis involves bacterial, viral, and fungal infections, toxicosis, and neoplasia [10]. The common diagnostic methods for confirmation of FB ingestion are radiography, proventricular endoscopy, and exploratory surgery [2]. In small birds, diagnostic imaging in the form of general or contrast radiography is especially important for such confirmation [9].

Endoscopic FB removal can be performed via two approaches: endoscopy alone as a non-invasive approach [1] or a combination of endoscopy and an invasive procedure, such as ingluviotomy [11], proventriculotomy [5], or ventriculotomy [4]. Endoscopic removal of FBs is a safe and minimally invasive procedure compared to the combined approaches [8]. Because of its simplicity, endoscopy is already commonly used in dogs and cats for the removal of FBs, whereas its use in avian species is restricted because of their different anatomical characteristics and variations in size.

The aim of this report was to present the outcomes of different patient positions and anesthesia methods for parrots during GI endoscopy for FB removal based on three retrospective cases.

## 2. Case Description

### 2.1. Case 1

A 1-year-old female blue-and-yellow macaw (*Ara ararauna*) weighing 812 g was presented to Chungbuk National University Veterinary Teaching Hospital 1 week after FB ingestion, necessitating medical intervention. The bird did not exhibit any GI symptoms, such as decreased appetite or vomiting. Physical examination revealed no consequential abnormalities. However, FBs were identified in the proventriculus by using X-ray imaging (Figure 1a). The FBs consisted of a silicon tube and a stainless steel needle hub used for syringe feeding.

For endoscopic removal of the FBs, anesthesia was induced with 5% isoflurane (Terrell; Piramal Critical Care, Toronto, ON, Canada) in 100% oxygen and maintained with 2% isoflurane in 100% oxygen, delivered using a mask. After induction of anesthesia, the patient was positioned in dorsal recumbency with the neck adequately extended to facilitate the procedure. The mask was temporarily repositioned to cover the nares during insertion of the endoscope. The operator slowly inserted the scope (diameter: 4.2 mm; EC-3890LK; PENTAX Medical, Tokyo, Japan) and an endoscopy processor (EPK-1000; PENTAX Medical, Tokyo, Japan) into the proventriculus. The tip of the feeding tube was placed in the upper proventriculus, which revealed that the mucus was normal. The transparent feeding tube was successfully removed using grasping forceps under endoscopic guidance (Figure 1b). After removal of the first FB (Case 1-1), the patient suddenly experienced respiratory arrest, and further procedures were terminated. Once the anesthesia was discontinued, the patient’s respiratory status returned to normal.

The following day, another endoscopic procedure (Case 1-2) was performed to remove the remaining FB from the proventriculus. The patient was placed in ventral recumbency, and the same steps were carried out for successful endoscopic extraction of the syringe hub (Figure 1c).

### 2.2. Case 2

A 6-month-old male Triton cockatoo (*Cacatua galerita triton*) weighing 555 g was presented to Chungbuk National University Veterinary Teaching Hospital 5 days after FB ingestion. The bird had a decreased appetite, but no other abnormalities were detected during the physical examination. X-ray imaging revealed an FB located in the proventriculus (Figure 2a). The FB consisted of a silicon tube.

For endoscopic removal of the FB, anesthesia was administered using the same protocol as that in Case 1, except that a 2.5mm uncuffed endotracheal tube was used instead of a mask. After induction of anesthesia, the patient was positioned in dorsal recumbency, and the same procedure was conducted as that in Case 1. Using grasping forceps, the transparent feeding tube was carefully secured and removed under endoscopic guidance (Figure 2b).

### 2.3. Case 3

A 4-month-old female blue-and-yellow macaw (*A. ararauna*) weighing 750 g was presented to Chungbuk National University Veterinary Teaching Hospital 1 day after FB ingestion. The bird exhibited lethargy, decreased appetite, and continuous vomiting, which had a green appearance. At a local veterinary clinic, X-ray imaging confirmed the presence of FBs in the proventriculus (Figure 3a).

For endoscopic removal of the FBs, anesthesia was administered using the same protocol as that in Case 1. After induction of anesthesia, the patient was positioned in ventral recumbency, and the same procedure was performed as that in Case 1. The mucus in the proventriculus was normal. The feeding tube, which was attached to a stainless steel syringe hub, was successfully extracted using grasping forceps during the endoscopic procedure (Figure 3b).

In the endoscopic view, the proventricular mucosa appeared normal in Cases 1 and 3, while it exhibited a greenish discoloration due to FB-induced irritation in Case 2 (Figure 4).

We evaluated the impact of the differences in endoscopic removal of the four FBs by summarizing the patient’s position during the procedure, anesthesia method (mask or intubation), operation times, vital signs, and range of saturation of percutaneous oxygen (SpO_2_) among the cases in Table 1.

## 3. Discussion

The primary cause of FB ingestion in this case series was the owners’ inexperience with hand feeding. High viscosity in the feeding formula increases the pressure within the syringe; thus, the syringe may easily detach from the feeding tube, causing it to enter the parrot’s mouth or crop during feeding.

In Cases 1 and 2, no clinical signs were evident except for a general poor condition, whereas Case 3 presented with symptoms of lethargy, reduced appetite, and persistent vomiting. In Case 3, a 15 cm undivided feeding tube was lodged between the parrot’s esophagus and proventriculus, causing incomplete proventriculus constriction and persistent vomiting.

For removal of FBs from the GI tract, an oral approach is suitable for smaller parrots (<400–500 g), whereas ingluviotomy can also be used for larger parrots (>400–500 g) [8]. However, we attempted endoscopic approaches in all cases, even though all of the patients weighed >500 g (812 g, 555 g, and 750 g), for the following reasons. First, only one or two FBs had to be removed in each case. Additionally, the endoscopic retrieval of blunt FBs poses a low risk of injury. Lastly, ingluviotomy can increase the risk of complications, such as leakage at the incision site, peritonitis, and dehiscence [4]. Consequently, if a small number of blunt FBs need to be removed, endoscopic removal is considered the safest procedure and is more appealing to owners than surgical intervention, regardless of the size of the parrot.

The rigid endoscope is a commonly employed instrument in ingluviotomy [8] and proventriculotomy [2], playing a crucial role in identifying and extracting foreign materials. In our cases, a flexible endoscope was more suitable for safely accessing the proventriculus through the oral cavity [6], minimizing the risk of crop wall perforation. The flexible endoscope proved to be applicable not only in larger birds (>1–2 kg) but also in those weighing less than 1 kg, based on our cases [8].

In our series, all of the endoscopic approaches were successful, and no severe complications occurred. Although respiratory arrest occurred in Case 1-1 owing to prolonged operation time, the FBs were safely removed via staged endoscopy over 2 days (Case 1-2). The patient in Case 2 experienced postoperative coughing for a week owing to endotracheal intubation. Coughing after extubation is a common complication in dogs and cats [12]. Similarly, endotracheal intubation in birds may induce upper respiratory tract irritation and subsequent coughing. In our cases, anesthesia was administered via a mask or endotracheal intubation, considering the small number of FBs and the potential for a quick operation. Although the masks covered only the patients’ nares during the endoscopic procedure, the anesthetic status remained stable in Cases 1-2 and 3 owing to the short operation time. In Case 1-1, apnea occurred due to the prolonged procedure without ventilation, highlighting the risks associated with maintaining anesthesia via a mask. If many FBs are present and stable anesthesia is required over a longer period, intubation via the air sac should be performed for safe endoscopic approach.

No marked differences in the induction or recovery times were observed across the four cases. However, anesthesia and operation times differed considerably. For Cases 1-1 and 2, conducted with the patient in dorsal recumbency, the endoscopic procedures took 30 and 15 min, respectively. In contrast, Cases 1-2 and 3, performed with the patient in ventral recumbency, required only 5 and 3 min, respectively. This highlights the fact that endoscopic procedures with the patient in dorsal recumbency take 5 to 10 times longer than those with the patient in ventral recumbency. Furthermore, in Cases 1-2 and 3, the patients’ SpO_2_ levels remained stable at 98–99%. In Cases 1-1 and 2, the patients’ SpO_2_ levels had a wider range (91–98%). These SpO_2_ level differences in patient positioning may be attributed to avian anatomical structures. In dorsal recumbency, the normal ventilation of a bird decreases owing to the compression of the abdominal and caudal thoracic air sacs by the weight of the abdominal viscera, which reduces the effective volume of the air sacs [13].

Overall, based on the analysis of three cases, ventral recumbency may result in a shorter endoscopic operation time and provide more stable SpO_2_ levels than dorsal recumbency during FB removal procedure. Regarding the anesthesia method, using a mask may be a preferable choice over endotracheal intubation, especially in straightforward and brief endoscopic procedures. Endotracheal intubation involves a delicate endoscopic procedure, and intubation via the air sacs is invasive, leading to a longer operation time. However, if the operation time exceeds the anticipated duration, it is advisable to consider switching to intubation to mitigate the risk of respiratory arrest in the patient. Therefore, the choice between employing a mask or endotracheal intubation should be made carefully, taking into account the side effects of each anesthesia method.

Our report retrospectively examines only three cases involving different patient positions and anesthesia methods, making it challenging to compare and draw conclusive findings regarding the optimal approach for GI endoscopy in FB removal. A greater number of cases would be required to validate the optimal patient positioning and anesthesia method. However, our cases provide insights that utilizing a mask or endotracheal intubation for anesthesia and positioning the patient in ventral recumbency may be beneficial for psittacine birds with one or two blunt FBs, particularly in the context of short-duration GI endoscopy. Staged endoscopy, as demonstrated in Case 1, can also be useful when the removal of FBs in one session is challenging [6]. Failure to access the ventriculus via the oral cavity may necessitate ingluviotomy or ventriculotomy [6,8]; however, endoscopy should first be attempted as it poses a lower risk of trauma or contamination than surgical approaches.

We recommend that a minimum of three individuals be present during the endoscopic procedure, including an endoscopist, a person monitoring anesthesia, and an assistant. Moreover, the endoscopist must be experienced to ensure safe execution. Unlike dogs and cats, birds have delicate crop walls, necessitating careful endoscopic access via the oral route to prevent perforations from extending from the crop into the air sacs.

## 4. Conclusions

Endoscopy is a safe and useful approach to remove a few blunt FBs from psittacine birds. This series demonstrated successful endoscopic removal of all FBs, with minimal complications. In terms of the anesthesia method and operational position, employing a mask with the patient in ventral recumbency may be considered when dealing with one or two blunt FBs, anticipating a short operation time. If operation time is expected to be prolonged, using intubation is more recommended than using a mask. Our series offers valuable insights into the safe removal of FBs in avian patients, along with practical guidance.

## Figures and Tables

**Figure 1 animals-13-03839-f001:**
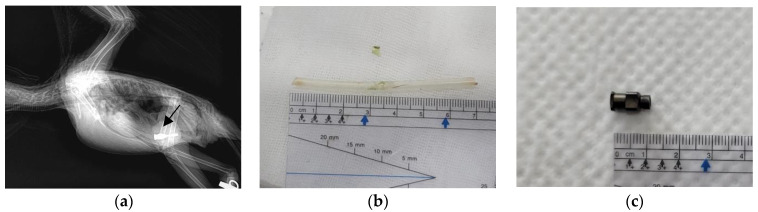
Images of the blue-and-yellow macaw (*Ara ararauna*) in Case 1. (**a**) X-ray image captured in the lateral view, clearly showing the presence of stainless steel foreign bodies (arrow). (**b**) Feeding tube removed under endoscopic guidance with the patient in ventral recumbency (Case 1-1). (**c**) Syringe hub removed under endoscopic guidance with the patient in dorsal recumbency (Case 1-2).

**Figure 2 animals-13-03839-f002:**
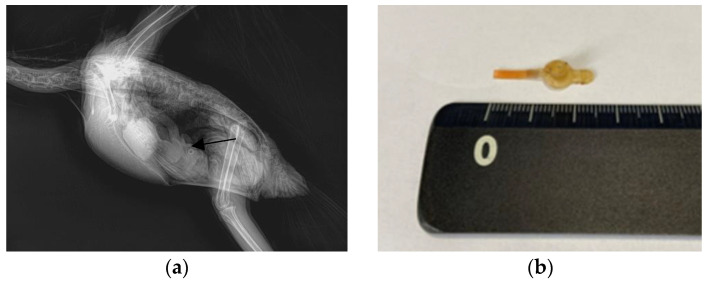
Images of the Triton cockatoo (*Cacatua galerita triton*) in Case 2. (**a**) A circular foreign body in the proventriculus, captured in an X-ray image in the lateral view (arrow). (**b**) The extracted silicon tube.

**Figure 3 animals-13-03839-f003:**
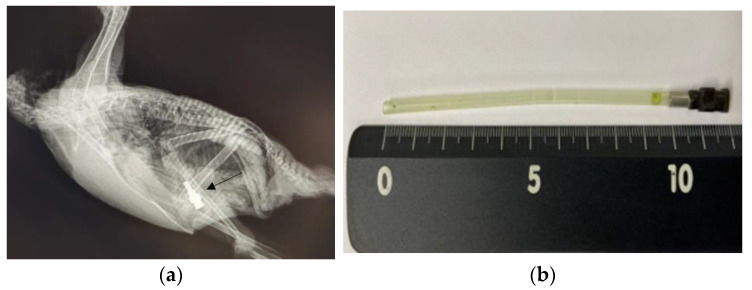
Images of the blue-and-yellow macaw (*A. ararauna*) in Case 3. (**a**) X-ray image in a lateral view, clearly showing the presence of stainless steel foreign bodies (arrow). (**b**) The extracted silicon tube and syringe hub.

**Figure 4 animals-13-03839-f004:**
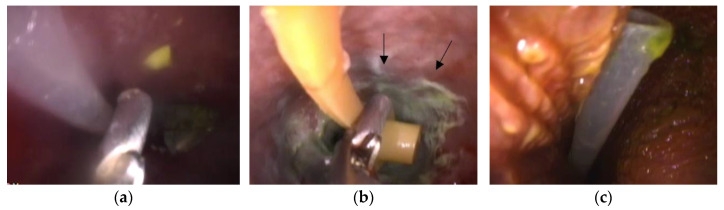
Images of the foreign body in the endoscopic view. (**a**) Grasping forceps and the feeding tube (Case 1-1). (**b**) Grasping forceps and the feeding tube (Case 2). White to greenish inflammatory plaques are observed on the proventricular mucosa (arrows). (**c**) The feeding tube (Case 3).

**Table 1 animals-13-03839-t001:** Characteristics of the four endoscopic procedures.

	Case 1-1	Case 1-2	Case 2	Case 3
Position	Dorsal recumbency	Ventral recumbency	Dorsal recumbency	Ventral recumbency
Anesthesia method	Mask	Mask	Intubation	Mask
Total operation time (min)	53	40	32	16
Induction time (min)	5	5	5	8
Anesthesia time (min)	43	30	22	5
Endoscopic operation time (min)	30	5	15	3
Anesthesia recovery time (min)	5	5	5	3
Vital signs before anesthesia				
Heart rate (beats/min)	240	240	168	180
Respiratory rate (breaths/min)	24	24	24	30
Vital signs during anesthesia				
Heart rate (beats/min)	241–258	240–264	150–240	150
Respiratory rate (breaths/min)	28–24	18–30	18–24	24
SpO_2_ during anesthesia (%)	93–98	98–99	91–98	98–99

Total operation time: the sum of induction time, anesthesia time, and recovery time; induction time: the duration from the initiation of anesthetic agent administration until the patient entered an unconscious state; anesthesia time: the period during which a patient remained in a state of anesthesia; endoscopic operation time: the interval between the initiation and completion of the endoscopic procedure; anesthesia recovery time: the duration from the cessation of anesthetic administration to the patient regaining consciousness; SpO_2_, saturation of percutaneous oxygen.

## Data Availability

The data presented in this study are available on request from the corresponding author.

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
