# Peer review of "Various Endoscopic Approaches for Removal of Proventricular Foreign Bodies in Parrots—Three Case Reports"

_animals, 2023, doi:10.3390/ani13243839_

Round 1
Reviewer 1 Report
Comments and Suggestions for Authors
The article presents 3 case reports but there is much improvement needed.
Please correct:
line 3- the title should be changed to 'three case reports'
My major concerns regard the discussions. The authors can not compare 3 mask inductions and maintenance with only 1 intubation, first of all. Then, to recommend in birds only mask anesthesia for maintenance while leaking Isoflurane and not being able to ventilate (see the first case during the first procedure and it's apnea) is a big problem. If the authors chose to only use Isoflurane and mask it is their decision, but they can NOT conclude that mask is better than intubation!
The quality of the pictures is a problem ( please remove the hands from the Xray of case 3 - where also the positioning of the patient is not correct) and there is only 1 endoscopic image.
There are no discussions regarding other endoscopic techniques, rigid ones, via ingluviotomy, since there are other articles about endoscopic removal of FB in birds.
The authors should stick to presenting their cases and not base conclusions on three cases, even if 1 of the cases had been anesthetized twice. Maybe it is more logical that the ventral approach is better since it does not cause that much compression on the air sacs but nevertheless the conclusion of ventral and dorsal should be based on a bigger nr of animals.
Author Response
Dear reviewer,
We are deeply grateful to the Assistant Editor (Valeriia Yurchenko) and the reviewers for their insightful comments on how to improve our paper entitled “Various Endoscopic Removal Approaches for Proventricular Foreign Bodies in Parrots–A Case Report”. We have revised the manuscript to incorporate their suggestions as follows and the manuscript includes revisions for other reviewers and assistant editor. Revised sentences and words were shown by red (for reviewer #1), blue (for reviewer #2), green (for reviewer #3) on the manuscript.
Please find the specific responses to each comment below. The complete responses to all the reviewers can be found in the attached file.
Comment 1: line 3- the title should be changed to 'three case reports'
Author response: We have revised it.
Comment2: The authors cannot compare 3 mask inductions and maintenance with only 1 intubation, first of all.
Author response: We agree with the reviewer’s opinion, so we have tried to revise the overall sentences in the manuscript.
- In L57-59, we removed the word "compare" and revised the sentence to clarify that this case report aims to present the outcomes of three cases involving different patient positions and anesthesia methods.
- In L205-207, we acknowledge that our findings do not allow us to conclusively determine which position and anesthesia method are optimal for the endoscopic procedure.
Comment3: Then, to recommend in birds only mask anesthesia for maintenance while leaking Isoflurane and not being able to ventilate (see the first case during the first procedure and it's apnea) is a big problem. If the authors chose to only use Isoflurane and mask it is their decision, but they can NOT conclude that mask is better than intubation!
Author response: As the reviewer pointed out, we cannot straightforwardly conclude that using a mask is superior to intubation based on the three cases. Therefore, we revised several sentences in the manuscript.
- In L15&L26-27, we added the condition that “within a short operation time” the mask may be effective in the endoscopic removal approach.
- In L178-181, we distinguished case 1-1 from cases 1-2 and 3 to emphasize the potential risks associated with using a mask as an anesthesia method.
- In L199-204, we attempted to emphasize that the use of mask is recommended primarily for straightforward brief endoscopic procedures, suggesting intubation for longer operation times.
Comment 4: The quality of the pictures is a problem (please remove the hands from the Xray of case 3 - where also the positioning of the patient is not correct) and there is only 1 endoscopic image.
Author response: We have replaced the X-ray image of case 3 with one in the appropriate position. Unfortunately, the resolution of this picture is the best quality available.
Concerning the endoscopic images, we have included one for each case and designated them as figure 4, with corresponding descriptions in L127-133. Please kindly understand that the quality of the pictures is not high, as the devise we used for endoscopic procedures was an older version.
Comment 5: There are no discussions regarding other endoscopic techniques, rigid ones, via ingluviotomy, since there are other articles about endoscopic removal of FB in birds.
Author response: We have incorporated a description of the rigid endoscopic technique in response to the reviewer’s comments in L164-169.
Comment 6: The authors should stick to presenting their cases and not base conclusions on three cases, even if 1 of the cases had been anesthetized twice.
Author response: We have attempted to adjust the overall tone of the manuscript to avoid making definitive statements.
- In L13-14, we have changed “the most effective approach” to “potentially effective approach”.
- In L20, we have replaced “focusing on optimal” to “employing different”.
- In L57-59, we have revised the aim of the report as mentioned above.
- In L207-212, we have stated that our cases offer limited evidence concerning anesthesia method and patient positioning.
- In L225-227, we have mentioned that “mask and ventral recumbency may be considered when dealing with one or two blunt FBs, anticipating a short operation time”.
Comment 7: Maybe it is more logical that the ventral approach is better since it does not cause that much compression on the air sacs but nevertheless the conclusion of ventral and dorsal should be based on a bigger nr of animals.
Author response: We agree with the reviewer’s opinion, so we have revised several sentences in the manuscript.
- In L27-28, we have added the limitation of our case reports in the abstract as you mentioned.
- In L205-207, as mentioned earlier, we acknowledge that our findings do not allow us to definitively conclude which position and anesthesia method are optimal for the endoscopic procedure.
Thank you for valuable review of our manuscript. We hope this paper is now suitable for publication in your journal.
Sincerely,
Dong-Hyuk Jeong
College of Veterinary Medicine, Chungbuk National University
Chungdaero-1, Cheongju, 28644, Republic of Korea
00821047492061
africabear@cbnu.ac.kr

Reviewer 2 Report
Comments and Suggestions for Authors
In the article "Various Endoscopic Removal Approaches for Proventricular 2 Foreign Bodies in Parrots–A Case Report", the Authors present three cases of proventricular foreign body removal via endoscopy in psittacines. The case presentation is clear, the images need a little improvement, and the discussion is sufficient. I like the comparison between intubation and mask anesthesia. I also have better outcomes using mask while dealing with such patients.
Please use the full name of the species in the figure descriptions.
The figures 1A, 2A and 3A would be much more valuable with increased size.
Author Response
Dear reviewer,
We are deeply grateful to the Assistant Editor (Valeriia Yurchenko) and the reviewers for their insightful comments on how to improve our paper entitled “Various Endoscopic Removal Approaches for Proventricular Foreign Bodies in Parrots–A Case Report”. We have revised the manuscript to incorporate their suggestions as follows and the manuscript includes revisions for other reviewers and assistant editor. Revised sentences and words were shown by red (for reviewer #1), blue (for reviewer #2), green (for reviewer #3) on the manuscript.
Please find the specific responses to each comment below. The complete responses to all the reviewers can be found in the attached file
In the article "Various Endoscopic Removal Approaches for Proventricular 2 Foreign Bodies in Parrots–A Case Report", the Authors present three cases of proventricular foreign body removal via endoscopy in psittacines. The case presentation is clear, the images need a little improvement, and the discussion is sufficient. I like the comparison between intubation and mask anesthesia. I also have better outcomes using mask while dealing with such patients.
Author response: Thank you for the thoughtful comment. Below are the specific responses to each comment.
Comment1: Please use the full name of the species in the figure descriptions.
Author response: We have revised it in each figure.
Comment2: The figures 1A, 2A and 3A would be much more valuable with increased size.
Author response: We have replaced the figures 1A and 2A with better quality. However, unfortunately, the resolution of figure 3A is the best quality available.
Thank you for valuable review of our manuscript. We hope this paper is now suitable for publication in your journal.
Sincerely,
Dong-Hyuk Jeong
College of Veterinary Medicine, Chungbuk National University
Chungdaero-1, Cheongju, 28644, Republic of Korea
00821047492061
africabear@cbnu.ac.kr

Reviewer 3 Report
Comments and Suggestions for Authors
Dear authors,
you presented three cases of removal of foreign bodies (feeding tubes) from three individual birds. Case reports are short, specific and give insight into whole process, from initial examination of the birds to the end of recovery. As negative, it could be described the small number of cases to be able to have more relevant results. The cases are not giving almost any new information regarding foreign bodies, parrots and endoscopy, maybe only related to the position of the birds body during the procedure; but, on the other hand, the articles dealing with this topic are not that common and could be valuable for small group of veterinarians working with such a patients in their clinics.
Specific comments:
Title - maybe consider to rephrase, as "Various Endoscopic Approaches for Removal of Proventricular Foreign Bodies in Parrots–A Case Report" or similar
L 14 "with a mask" - please rephrase this sentence so it is clear that you mean anesthesia via mask
L 38 please rephrase "materials in the house"
Author Response
Dear reviewer,
We are deeply grateful to the Assistant Editor (Valeriia Yurchenko) and the reviewers for their insightful comments on how to improve our paper entitled “Various Endoscopic Removal Approaches for Proventricular Foreign Bodies in Parrots–A Case Report”. We have revised the manuscript to incorporate their suggestions as follows and the manuscript includes revisions for other reviewers and assistant editor. Revised sentences and words were shown by red (for reviewer #1), blue (for reviewer #2), green (for reviewer #3) on the manuscript.
Please find the specific responses to each comment below. The complete responses to all the reviewers can be found in the attached file
you presented three cases of removal of foreign bodies (feeding tubes) from three individual birds. Case reports are short, specific and give insight into whole process, from initial examination of the birds to the end of recovery. As negative, it could be described the small number of cases to be able to have more relevant results. The cases are not giving almost any new information regarding foreign bodies, parrots and endoscopy, maybe only related to the position of the birds body during the procedure; but, on the other hand, the articles dealing with this topic are not that common and could be valuable for small group of veterinarians working with such a patients in their clinics.
Author response: Thank you for your considerate review. Below are the specific responses to each comment.
Comment 1: Title - maybe consider to rephrase, as "Various Endoscopic Approaches for Removal of Proventricular Foreign Bodies in Parrots–A Case Report" or similar
Author response: We have revised the title, considering the reviewers' comments.
Comment 2: L 14 "with a mask" - please rephrase this sentence so it is clear that you mean anesthesia via mask
Author response: We have rephrased it.
Comment 3: L 38 please rephrase "materials in the house"
Author response: We have rephrased it to “household items”.
Thank you for valuable review of our manuscript. We hope this paper is now suitable for publication in your journal.
Sincerely,
Dong-Hyuk Jeong
College of Veterinary Medicine, Chungbuk National University
Chungdaero-1, Cheongju, 28644, Republic of Korea
00821047492061
africabear@cbnu.ac.kr

Round 2
Reviewer 1 Report
Comments and Suggestions for Authors
Dear authors, thank you for the changes, I believe the article is improved.
I still would like to emphasize on the mask vs intubation - see line 26; I do not see any reason why a mask would be preferred over ET intubation. Please refer from using this sentence or idea throughout the text.
Line 57 - change 'to show' to 'to present' or 'to describe'
Author Response
Dear reviewer,
We are deeply grateful to the Assistant Editor (Valeriia Yurchenko) and the reviewers for their insightful comments on how to improve our paper entitled “Various Endoscopic Approaches for Removal of Proventricular Foreign Bodies in Parrots–Three Case Reports”. We have revised the manuscript for the minor revision to incorporate the suggestions as follows and the manuscript includes revisions for the reviewer #1. Revised sentences and words were shown by red (for reviewer #1) on the manuscript. Our previous manuscript ID was animals-2681802.
Please find the specific responses to each comment below. The complete responses to all the reviewers can be found in the attached file.
COMMENTS FROM REVIEWER #1
Dear authors, thank you for the changes, I believe the article is improved.
Comment 1: I still would like to emphasize on the mask vs intubation - see line 26; I do not see any reason why a mask would be preferred over ET intubation. Please refer from using this sentence or idea throughout the text.
Author response: Throughout our case report, we mentioned that a mask may be preferred over endotracheal intubation in very simple and short operation times, considering its practical benefits from our cumulative clinical experiences. However, as the reviewer pointed out, it is advisable not to definitively state a preference for a mask based on three cases. Consequently, we have made several revisions in our manuscript as follows.
- L14-15: We have added ‘or endotracheal intubation’ to avoid implying that mask is superior to ET intubation.
- L26-27: We have revised the sentence to provide options to the reader, suggesting either employing a mask or ET intubation based on the anticipated operation time.
- L205-206: We have added a sentence to emphasize that the choice between mask and an ET tube for anesthesia should be made carefully.
- L 211-212: We have added ‘or endotracheal intubation’ and revised the sentence to convey the insights from our report.
- L 229-230: We have added a condition when ET intubation is recommended.
Comment 2: Line 57 - change 'to show' to 'to present' or 'to describe'
Author response: We have revised it ‘to present’
Thank you for valuable review of our manuscript. We hope this paper is now suitable for publication in your journal.

Reviewer 3 Report
Comments and Suggestions for Authors
Dear authors, thank you for taking into account all the comments given by the reviewer.
Best regards
Comments on the Quality of English Language-
Author Response
Thank you so much for your comment.
We have revised the manuscript for the minor revision to incorporate the suggestions as follows and the manuscript includes revisions for the reviewer #1. Revised sentences and words were shown by red (for reviewer #1) on the manuscript. Our previous manuscript ID was animals-2681802.
The complete responses to all the reviewers can be found in the attached file.

Round 3
Reviewer 1 Report
Comments and Suggestions for Authors
Even though the information is not new, it does provide clinical insights on the subject.